# Towards Improved Management of Tuberculous Bloodstream Infections: Pharmacokinetic Considerations with Suggestions for Better Treatment Outcomes

**DOI:** 10.3390/antibiotics11070895

**Published:** 2022-07-05

**Authors:** Charles Okot Odongo, Lydia Nakiyingi, Clovis Gatete Nkeramihigo, Daniel Seifu, Kuteesa Ronald Bisaso

**Affiliations:** 1Division of Basic Medical Sciences, School of Medicine, University of Global Health Equity, Kigali P.O. Box 6955, Rwanda; clovis.nkeramihigo@student.ughe.org (C.G.N.); dseifu@ughe.org (D.S.); 2Department of Internal Medicine, School of Medicine, College of Health Sciences, Makerere University, Kampala P.O. Box 7072, Uganda; lydiakiyingi@gmail.com; 3Breakthrough Analytics Ltd., Plot 17B Gasper Oda Close, Naguru, Kampala, Uganda; kuteesar@gmail.com

**Keywords:** tuberculosis, sepsis, treatment, pharmacokinetics, C_max_, AUC

## Abstract

*Mycobacterium tuberculosis* is the leading cause of sepsis among HIV-infected adults, yet effective treatment remains a challenge. Efficacy of antituberculous drugs is optimized by high Area Under Curve to Minimum Inhibitory Concentration (AUC/MIC) ratios, suggesting that both the drug concentration at the disease site and time above MIC are critical to treatment outcomes. We elaborate on sepsis pathophysiology and show how it adversely affects antituberculous drug kinetics. Expanding distribution volumes secondary to an increased vascular permeability prevents the attainment of target C_max_ concentrations for nearly all drugs. Furthermore, sepsis-induced metabolic acidosis promotes protonation, which increases renal clearance of basic drugs such as isoniazid and ethambutol, and hence AUCs are substantially reduced. Compared with the treatment of non-sepsis TB disease, these distorted kinetics underlie the poor treatment outcomes observed with bloodstream infections. In addition to aggressive hemodynamic management, an increase in both the dose and frequency of drug administration are warranted, at least in the early phase of treatment.

## 1. Introduction

In regions of the world with high HIV burdens, mycobacteria have become the leading cause of sepsis among adults, accounting for up to 40% of bloodstream infections [1,2,3,4]. Compared to non-sepsis tuberculous disease, bloodstream infections are associated with a 2.5-fold increased risk of mortality [5]. Up to 85% of all patients with tuberculous bloodstream infections are HIV co-infected, and mortality in such patients may be as high as 50% [6,7,8,9,10]. From the literature, such grim statistics appear to be driven by several factors that continue to hinder the effective management of tuberculous bloodstream infections. First, unlike pulmonary disease, bloodstream infections tend to be insidious, often with atypical presentation, and most patients rarely expectorate. This leads to a low index of suspicion among clinicians [7,11]. Secondly, even where such infection is suspected, most settings lack the tools for an effective and timely diagnosis, and hence, treatment tends to be delayed, usually considered after a failure to improve on conventional antibiotics. With recent advances in diagnostics, novel tools with short turnaround times such as GeneXpert & Urine TB-LAM have improved diagnosis and contributed to the growing awareness of the high burden of tuberculous bloodstream infections. However, despite these advances, the trend in recent studies suggests that the management of such infections remains a challenge for additional reasons [9,12]. In this submission, we posit that the complex pharmacokinetic changes induced by sepsis pathophysiology are perhaps most critical, and that unless clinicians proactively account for these changes, attempts at effective treatment of tuberculous sepsis will foreseeably remain challenging.

Whereas the adverse outcomes of tuberculous bloodstream infections are well described, few studies have attempted to correlate these outcomes with actual drug exposure. To argue our case, we draw insights from a recent study that attempts to address this question [12]. In this study, Rao et. al. report the pharmacokinetics of anti-tuberculous drugs among patients with tuberculous bloodstream infections with HIV co-infection. Of the 81 participants enrolled and started on the standard first-line regimen, close to 40% were unable to return for pharmacokinetic studies that were scheduled two weeks later. A breakdown of this figure showed that 22% had died, 6% had withdrawn from the study and 10% were presumably lost to follow up. Among those successfully studied (60%), significantly low plasma drug levels were reported for all drugs with the majority failing to reach two or more desired pharmacokinetic targets. Isoniazid showed the lowest serum concentrations with only 4% of participants achieving target exposure (AUC_0–24_) despite dosing using weight-based bands, designed to optimize drug exposure. The best pharmacokinetic profile was seen with pyrazinamide where 88% of participants achieved the target C_max_ concentrations. Even then, only 39% achieved the desired target exposure (AUC_0–24_). The study observed wide variabilities in pharmacokinetic parameters, which remain unexplained after investigating several covariates. These findings are summarized in Table 1. To set the stage for interpreting these findings, a brief review of some pharmacokinetic concepts as well as the pathophysiology of sepsis are warranted.

## 2. Review of Pharmacokinetic Concepts and Terminologies

Pharmacokinetics describes the fate of a drug in the body. It involves characterizing the rate and extent to which a drug can be absorbed, distributed, metabolized and/or eliminated from the body. These processes are quantitatively captured by primary pharmacokinetic parameters such as the absorption rate constant (*k_a_*), volume of distribution (V_d_) and clearance (CL). Whereas *k_a_* is self-explanatory, V_d_ refers to the apparent volume in which the drug has to be contained in order to provide the same concentration as that observed in plasma. Clearance, on the other hand, quantifies the removal of a drug from the body via processes commonly involving first-order kinetics (normally metabolic conversion or renal excretion). Central to the determination of these parameters through compartmental or non-compartmental analyses is the plasma drug concentration versus time profile, which provides a graphical summary of the fate of the drug in the body. The analysis of such profiles enables the determination of secondary pharmacokinetic parameters such as the peak plasma concentration (C_max_), half-life (t ½), and area under the plasma concentration versus time curve (AUC), which together summarize the overall rate and extent of exposure of the body to the drug. These are often referred to as exposure parameters. Whereas primary pharmacokinetic parameters are critical to understanding the influence of pathophysiology on the fate of a drug, secondary parameters are of a more practical importance to the clinician. Nevertheless, secondary parameters are related to primary parameters, from which they can be mathematically derived. For instance, C_max_ is directly proportional to *k_a_* and decreases when V_d_ and CL increase, while AUC is directly related to the drug dose and inversely related to CL. Furthermore, a drug’s half-life increases with V_d_ and decreases with CL. As a result of these relationships, differential changes in the exposure parameters provide a clue for the practicing clinician about which specific pharmacokinetic process has been altered [13].

## 3. Pathophysiology of Sepsis

It is generally agreed that, regardless of the underlying pathogen, sepsis appears to have a similar clinical phenotype [14]. Early on, endotoxins trigger the release of inflammatory mediators (e.g., prostaglandins, leukotrienes, and cytokines), which reduce the peripheral vascular tone in both arterial and venous circulations. This decreases tissue perfusion and promotes anaerobic metabolism with lactic acidosis. Additionally, capillary beds dilate, causing an increased leakage of plasma and inflammatory cells into the interstitium. These changes, in principle, expand the volume of distribution (V_d_) of nearly all drugs, although the greatest effect is likely seen on drugs with an initially low V_d_ [15]. Clinical findings in early sepsis include fever, tachycardia, tachypnea, respiratory alkalosis, and an increased cardiac output, often with warm dry extremities. If left unchecked, the risk of homeostatic dysregulation increases as marked acidemia progressively triggers varying degrees of myocardial and respiratory dysfunction [16]. In the absence of fluid resuscitation, cardiac output decreases, and patients run cold extremities as well as appearing clammy, blunt, and often confused. Eventually, such patients become comatose and die. The precise pattern and duration of these events varies with each case depending on the tempo of infection, patient age, presence of comorbidities or therapeutic interventions instituted. Therefore, as a clinical entity, sepsis is a serious condition that warrants aggressive hemodynamic and antimicrobial treatment in which time is of the essence.

That participants in the above study were discharged from the hospital to continue with home medication attests to some level of hemodynamic stability. However, despite a considerable effort to dose appropriately, the subsequent high deaths and loss to follow up, in our view, suggest that the therapeutic effort was grossly ineffective. This proposition is borne out by the consistently low exposure observed for nearly all drugs in this study. However, contrary to this trend, the authors speculated that deaths were likely due to higher (probably toxic) drug levels in the affected individuals. We therefore discuss the anticipated effects of sepsis pathophysiology on anti-tuberculous drug kinetics and show how this was critical to interpreting the results observed in the study.

## 4. Effect of Sepsis on Drug Distribution Volumes and Kinetics

Because sepsis pathophysiology increases capillary permeability, it would be expected that the tissue distribution (V_d_) of all anti-tuberculous drugs studied would increase. As such, a reduction in plasma concentrations as measured by C_max_ would be inevitable as more drug moves into the interstitium compared with what would be expected with a similar dosing in the non-sepsis state. Indeed, this trend was observed for all drugs in this study, as shown by the low proportions of participants who attained target C_max_ exposures (the relatively higher proportions for isoniazid & pyrazinamide will be explained later). Table 1 summarizes these data as extracted from Rao et al. [12]. For non-sepsis tuberculous disease, such an increased tissue drug distribution may be favorable, since more drug would likely reach extravascular disease sites. However, such a phenomenon would not be favorable in tuberculous sepsis, as drug concentrations are effectively decreased in plasma, the primary disease site. We therefore contend that such sepsis-induced expansion in drug distribution volumes was partly responsible for the failure to achieve target C_max_ concentrations for most participants, despite weight-based dosing. As shown in Table 1, only 8.2% & 30.6% of rifampin and ethambutol concentrations, respectively, reached the target C_max_ concentrations. Considering that C_max_ concentrations have a direct bearing on anti-tuberculous drug efficacies [17], the effectiveness of rifampin and ethambutol was most affected by this phenomenon.

## 5. Effect of Sepsis-Induced Acidemia on Drug Excretion Kinetics

In this study, over 63% of participants achieved target isoniazid C_max_ concentrations, and yet only 4% achieved target AUC_0–24_ exposures. In addition, genotyping revealed that more than 90% of the participants examined were slow-intermediate acetylators. Such a population would normally be expected to experience higher AUCs as less drug is cleared per unit time. Surprisingly, this was not the case given that only 4% reached the target AUC of 52 mg*h/L (Table 1). In the case of ethambutol, there was no reference data on its target AUC, and hence it was not possible to determine the proportion that reached target exposure values. Given that both isoniazid and ethambutol are weak bases (see structural representation in Figure 1), we contend that the protonated species of both drugs would predominate within the acidic milieu of sepsis. As such, once filtered into the renal tubular environment, these species would be poorly reabsorbed into the renal interstitium. In other words, the sepsis state effectively increased renal clearance (excretion) of these drugs beyond what would normally be expected under non-sepsis conditions. Indeed, this trend was clearly observed for isoniazid in this study, as shown by the low proportion of participants (4%) who attained the target AUC, even though more than 63% had initially attained target C_max_ concentrations (Table 1). Furthermore, we contend that such increased renal clearance would be particularly pronounced with ethambutol, since even without sepsis, over 80% of this drug is normally excreted unchanged (owing to its relatively hydrophilic nature) [17]. This premise is supported by the fact that despite being one of the highest dosed drugs (the mean dose administered was 18.4 mg/kg), ethambutol experienced the lowest exposure (AUC) of all drugs in the regimen, as shown in Table 1. In addition, because the free drug is generally subject to unrestricted renal filtration, it can be further argued that low protein binding indirectly promoted renal excretion of both ethambutol and isoniazid. This is generally supported by data from the literature suggesting that both drugs tend to be poorly protein-bound (Table 2). Interestingly, isoniazid is reported to have a protein binding close to zero, suggesting that it exists completely free and unbound in plasma. This would predictably make it highly susceptible to filtration kinetics, in addition to its protonated species not being easily reabsorbable under sepsis conditions. It is not surprising, therefore, that even with dose tripling (from 300 to 900 mg), simulation studies done by Rao et. al. predicted that only 4.5% of patients would reach the desired target concentrations.

In contrast to our observations on isoniazid and ethambutol, we contend that acidosis altered the excretion kinetics of pyrazinamide in the opposite direction. Pyrazinamide exhibited the most efficient absorption kinetics of all the first-line antituberculous drugs, as shown by the 88% who attained target C_max_ concentrations. This observation is consistent with previous literature showing the highest absorption rate constant of all (Table 2), which, coupled with a low pre-systemic metabolism, translates into a bioavailability normally exceeding 95% [17]. Furthermore, it has previously been reported that in acidic medium, the Henderson–Hasselbalch equilibrium progressively favors the formation of the unprotonated (unionized) species of pyrazinoic acid [18]. Even though low protein binding would comparatively increase its renal filtration, such a species, being more lipid-soluble, would nevertheless be easily reabsorbed from the acidic renal tubular environment. In effect, sepsis decreases the renal clearance of pyrazinamide. In our view, this explains the comparatively higher proportion (39%) of participants who achieved the target AUC, a feat that was unlike any of the other drugs studied under these conditions. Interestingly, since pyrazinoic acid is also the pharmacodynamically active drug species, we contend that sepsis pathophysiology would technically enhance its efficacy. In fact, Zhang et. al. showed that the drug achieved a better concentration inside mycobacteria at low pH, as this would predominantly favor the existence of the unprotonated form, enhancing influx [18]. Of interest, both Pasipanodya et al. [19] and Chideya et. al. [20] have previously reported on the critical role of pyrazinamide in predicting treatment outcomes. We believe these findings are explained by the favorable pharmacokinetic–pharmacodynamic changes described above.

## 6. Discussion

The phenomenon of sepsis-induced sub-optimal drug exposure has been previously described with beta-lactam [21,22] and aminoglycosides [23,24] antibiotics. A critical examination of anti-tuberculous drug kinetics under sepsis conditions was therefore long overdue, given the high disease burden and associated poor treatment outcomes in the literature. The rationale for combination drug use in TB treatment is twofold: to prevent the selection of drug resistance as well as to optimize drug synergies in order to enhance treatment efficacy. The current first-line regimen was developed more than 60 years ago, in the pre-HIV era when pulmonary TB was the predominant disease. Even though this regimen continues to work well for most TB diseases, sepsis clearly presents a unique scenario in which drug disposition appears to be completely distorted. The drug profiles described in Rao et al. reflect steady state concentrations when it was expected that plasma and tissue concentrations were parallel, and each drug was in the terminal log-linear phase of its concentration–time curve. Even though none of the 40% dropouts contributed data to this study, we contend that the poor drug profiles and trends reported here reflect what transpired in their bodies, if not worse. It is our view that few participants, if any, would die, withdraw from the study, or fail to turn up for medical review while receiving effective treatment and experiencing significant improvements. Therefore, the high incidences of deaths, withdrawals, and loss to follow up observed in this study speak volumes about the efficacy of the current treatment regimen under sepsis biology. As such, treatment optimization in tuberculous sepsis is a pharmacokinetic challenge that requires urgent attention. Our view is that drugs such as isoniazid and ethambutol, whose excretion is predominantly increased in sepsis, may need to be administered more frequently than is currently the case. Alternatively, such drugs may warrant dose loading or substitution with other drugs whose renal reabsorption would be predicted to increase under acidic conditions. Hence, the urgent need to explore these alternatives.

Several studies have previously demonstrated that the bactericidal and sterilizing effects of rifampin [19,25], isoniazid [19,26], pyrazinamide [19,27,28] and ethambutol [29] are all optimized by high AUC/MIC ratios. In other words, both the concentration at the disease site and time above MIC are critical for an optimal anti-tuberculous effect. However, given that sepsis expands the distribution volumes and increases the renal clearance of some drugs, it is impossible to optimize treatment under the current dosing scheme. This was clearly demonstrated by the simulation studies performed by Rao et al., and the case of isoniazid was particularly revealing. Had reference AUC values been available, we believe a similar picture would have emerged with ethambutol simulation studies. This study also highlights an important pharmacokinetic principle that clinicians ought to keep in mind: that whereas an increased tissue distribution may be desirable for antibiotics in many disease states, this does not hold true for sepsis, since it significantly transfers drugs away from the intravascular space, the primary site of drug action in bloodstream infections [15]. Furthermore, where combination treatment is employed, the picture is even more complex, as any potential synergies, including the ability of drugs to protect each other against resistance selection, tend to be lost by such distorted kinetics. Weiner et al. have demonstrated similar risks with intermittent rifabutin plus isoniazid in TB/HIV co-infected patients [30]. In the case of isoniazid, for instance, despite high absorption and C_max_ levels achieved by most participants in Rao et al., the overall exposure as measured by AUC_0–24_ was surprisingly dismal, yet this drug forms the backbone of the current TB treatment regimen. In view of these findings, both the dose and frequency of isoniazid and ethambutol administration may warrant increases, at least within the first week of treatment. We have embarked on simulation studies to explore alternative dosing schemes optimized to sepsis pharmacokinetics.

Lastly, due to the chronic nature of HIV illness, it is impossible to fully appreciate or account for the complex pathophysiological changes such patients undergo before or during intercurrent infections like tuberculosis. We contemplated the possibility of drug malabsorption contributing to the findings in Rao et al., given previous reports [31,32]. We are, however, confident that such a possibility was unlikely for the following reason: From the literature, pyrazinamide followed by isoniazid shows the best absorption and bioavailability parameters (Table 2). Interestingly, in Rao et al., the same drugs achieved the highest proportion of target C_max_ scores in that same order, i.e., 88% & 66.3% (Table 1). This, in our view, was unlikely to be a mere coincidence, but a reflection of the normal absorption pattern consistent with the literature. Nevertheless, we appreciate that patients with such chronic illnesses will often present with underlying organ abnormalities that may impact pharmacokinetics in many as yet unclear ways. For a long time, it was widely believed that patient factors (e.g., adherence), more than anything else, accounted for many of the poor outcomes of TB treatment. However, studies comparing treatment outcomes in self-administered and directly-observed therapy have shown this to be false [33,34,35]. Given this evidence and the benefit of hindsight, it is clear that challenges with TB treatment have for long been perceived and approached from a somewhat narrow perspective. In fact, research is increasingly revealing the critical role of pharmacokinetic factors in determining treatment outcomes [19,36,37]. The analysis and conclusions presented here are in line with this new paradigm. We hope that the viewpoints shared in this paper will activate interest and promote awareness of these challenges, especially among clinicians in the developing world where the dynamics of optimizing TB treatment are yet to be fully understood.

## Figures and Tables

**Figure 1 antibiotics-11-00895-f001:**
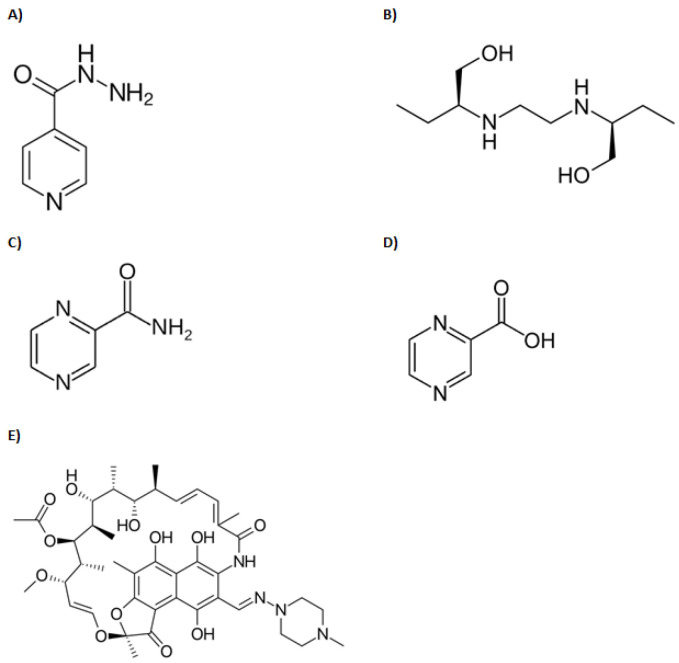
Structures of anti-tuberculous drugs reviewed in this manuscript: (**A**) Isoniazid, (**B**) Ethambutol, (**C**) Pyrazinamide, (**D**) Pyrazinoic acid, (**E**) Rifampin.

**Table 1 antibiotics-11-00895-t001:** Summary of antituberculous drug exposure parameters and desired targets as extracted from Rao et al. [12].

Drug	Dose (mg/kg)	C_max_ ^a^ (mg/L)	Target C_max_ (mg/L)	% Attained C_max_	AUC_0–24_ ^b^ (mg ×·h/L)	Target AUC_0–24_	% Attained target AUC_0–24_
	Median (IQR) ^c^	Median (IQR)			Median (IQR)		
Rifampin	10.1 (9–10.7)	3.8 (2.3–5.3)	≥8.0	8.2	21.7 (13.4–31.2)	≥35.4	16.3
Isoniazid	5.0 (4.5–5.4)	3.6 (2.3–4.6)	≥3.0	63.3	22.5 (14.3–34)	≥52.0	4.1
Pyrazinamide	25.4 (23–28)	34 (28.3–44)	≥20	87.8	351 (237.1–477.9)	≥363	38.8
Ethambutol	18.4 (16.5–19.6)	1.8 (1.3–2.2)	≥2.0	30.6	14.3 (10.6–26.6)	-	-

^a^ C_max_ = maximum plasma drug concentration, ^b^ AUC_0–24_ = area under the plasma concentration-time curve over 24 h, ^c^ IQR = interquartile range.

**Table 2 antibiotics-11-00895-t002:** Primary pharmacokinetic parameter estimates for antimycobacterial drugs in adult patients with non-sepsis tuberculous disease. Source: data extracted from [17].

Drug	*k_a_* (h^−1^)	Bioavailability (F)	Protein Binding (%)	Clearance (CL/F) (L/h)	V_d_ (L)
Rifampin ^a^	1.15	0.68	60–90	12.6 ^b^	58.2
Isoniazid	2.3	≈1.0	≈0	13.32 ^c^	40.2
Pyrazinamide	3.56	≈0.95	10	3.96	34.2
Ethambutol	0.7	0.77 (±8)	6–30	31	96

^a^—Active deacetyl metabolite, ^b^—CL/F is higher after repeated administration, ^c^—data presented here is for slow-intermediate acetylators only, *k_a_* = absorption rate constant, CL/F = clearance relative to bioavailability, V_d_ = apparent volume of distribution.

## Data Availability

Not applicable.

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
