# Peer review of "Towards Improved Management of Tuberculous Bloodstream Infections: Pharmacokinetic Considerations with Suggestions for Better Treatment Outcomes"

_antibiotics, 2022, doi:10.3390/antibiotics11070895_

Round 1
Reviewer 1 Report
The authors have adequately answered my questions regarding the limitation of just using one research paper to write the perspective.
As per my own observations, there is only one paper that looks at TB sepsis patients who are also HIV positive. Therefore, I do agree with the authors on the need of conducting more pharmacokinetic studies in HIV patients with TB sepsis.
Author Response
We sincerely thank the reviewer for positively receiving our responses regarding the questions raised previously. We also note that going by the check box below, there are no major issues raised, except for the fact that the reviewer felt that "moderate English changes were required", for which we respond as follows:
The request put before us is particularly challenging since the revisions required appear to be minor. We kindly request the reviewer to be more specific and point out the particular sections or paragraphs that may require revision/editing. We shall be happy to oblige to such specific items since we wish to preserve the rest of the manuscript in its current text.
Reviewer 2 Report
The manuscript presented for review touches on a key topic: treating mycobacterial tuberculosis sepsis. The authors emphasize that such therapy is very complex and demanding in the submitted text.
However, I still miss a clear indication of the population / region to which the article relates in the title or summary.
I maintain that, in my opinion, basing the work on more articles would increase its value, as relying on the data presented in this one work may be insufficient and perhaps a simplification. Of course, I understand that the assumptions of the perspective article do not preclude this approach; however, it would be much clearer to analyze the results also in the context of other works.
However, I appreciate that the authors referred to some of the comments presented in previous reviews.
Author Response
The manuscript presented for review touches on a key topic: treating mycobacterial tuberculosis sepsis. The authors emphasize that such therapy is very complex and demanding in the submitted text.
However, I still miss a clear indication of the population / region to which the article relates in the title or summary.
Response: We have responded to this comment previously. In brief, our article does not seek to limit itself to a particular population or geographic region. The topic addressed is relevant globally since both HIV and tuberculosis are epidemiologically global in nature. If this response seems insufficient, further guidance is sought from the editors on how best to respond.
I maintain that, in my opinion, basing the work on more articles would increase its value, as relying on the data presented in this one work may be insufficient and perhaps a simplification. Of course, I understand that the assumptions of the perspective article do not preclude this approach; however, it would be much clearer to analyze the results also in the context of other works.
Response: We have also provided a response to this comment in our previous submission. in brief, whereas we agree with the reviewer that other related studies could potentially enrich our manuscript and maybe even strengthen our conclusions, the reality is that no such studies exist in the literature. This fact has been acknowledged in lines 49-53 and there is nothing more we can do about it.
Additional general comment:
We have also used the services of a native English speaker to revise the entire manuscript. We hope the message is still preserved but now clearer than before.
This manuscript is a resubmission of an earlier submission. The following is a list of the peer review reports and author responses from that submission.
Round 1
Reviewer 1 Report
The manuscript presented to me for review touches on a crucial topic: the treatment of tuberculosis and, more specifically, sepsis caused by mycobacteria. The authors emphasize that such therapy is very complex and demanding in the presented text.
However, I miss a clear indication of the population/region the article relates to in the title or abstract. The authors of the article refer to the work of Rao et al. in which a study checking the phramakokinetics of the most common anti-tuberculosis drugs on a group of 81 people has been described in detail. In my opinion, basing the work only on the data presented in this work may be insufficient and maybe a simplification. Of course, I understand that the assumptions of the perspective article do not preclude such an approach; however, it would be much more readable to analyze the results in the context of other works as well. The more so because the manuscript's authors also refer to other works on pharmacokinetics in tuberculosis. Therefore, I believe that the work and analysis of the article by Ruo et al. should present the context of the research results of a larger number of researchers, which was strengthened by the overtones of the results.
Coming to the content of the manuscript itself, its structure is well thought out but not without shortcomings.
I suggest the authors recheck the manuscript for both spelling and punctuation.
The page numbering in the presented manuscript is also incorrect.
In line 52, I suggest the use of Ruo et al.
In line 175 and Table 1, please check the correct notation of units as there should be a multiplication sign.
The presentation of Figure 1 in the context of the presented manuscript seems unnecessary.
Reviewer 2 Report
Odongo and the authors have discussed the effect of sepsis on the biodistribution and bioavailability of antimicrobials for TB blood infection treatment in HIV-affected patients.
This perspective makes some solid points to consider about the challenges to attend the AUC of drug availability in blood and in infected tissues.
In one study (33875424), it has been proposed that individualized dosing could provide a solution to the issues raised in this perspective. The authors are requested to address whether individualized dosing could provide a solution to the drug availability in HIV patients suffering from sepsis.